Ecodatacube.eu: analysis-ready open environmental data cube for Europe

http://orcid.org/0000-0002-0962-6478 Witjes Martijn 1 martijn.witjes@opengeohub.org
http://orcid.org/0000-0003-1589-0467 Parente Leandro 1
http://orcid.org/0000-0002-4557-3537 Križan Josip 2
http://orcid.org/0000-0002-9921-5129 Hengl Tomislav 1
Antonić Luka 2
1 OpenGeoHub , Wageningen , Netherlands
2 MultiOne , Zagreb , Croatia
Wang Jingzhe
Electronic publication date: 2023 Jun 6
Publication date: 2023
Volume: 11
Electronic Location ID: e15478
Received 2022 Nov 22; Accepted 2023 May 8
Copyright: © 2023 Witjes et al.
Copyright year: 2023
Copyright holder: Witjes et al.
License: This is an open access article distributed under the terms of the Creative Commons Attribution License, which permits unrestricted use, distribution, reproduction and adaptation in any medium and for any purpose provided that it is properly attributed. For attribution, the original author(s), title, publication source (PeerJ) and either DOI or URL of the article must be cited.
License URL: https://creativecommons.org/licenses/by/4.0/

Keywords: Sentinel-2, Landsat, Data cube, Digital terrain model, Elevation, Gap filling, Analysis-ready data, Land cover, Lucas, EcoDataCube

Funding: European Union’s the Innovation and Networks Executive Agency (INEA) under Grant Agreement Connecting Europe Facility (CEF) Telecom Project 2018-EU-IA-0095 European Union’s Horizon Europe Research and Innovation Programme 101059548 This work has received funding from the European Union’s the Innovation and Networks Executive Agency (INEA) under Grant Agreement Connecting Europe Facility (CEF) Telecom project 2018-EU-IA-0095 and from the European Union’s Horizon Europe research and innovation programme under grant agreement No. 101059548. The funders had no role in study design, data collection and analysis, decision to publish, or preparation of the manuscript.

==============================
The article describes the production steps and accuracy assessment of an analysis-ready, open-access European data cube consisting of 2000–2020+ Landsat data, 2017–2021+ Sentinel-2 data and a 30 m resolution digital terrain model (DTM). The main purpose of the data cube is to make annual continental-scale spatiotemporal machine learning tasks accessible to a wider user base by providing a spatially and temporally consistent multidimensional feature space. This has required systematic spatiotemporal harmonization, efficient compression, and imputation of missing values. Sentinel-2 and Landsat reflectance values were aggregated into four quarterly averages approximating the four seasons common in Europe (winter, spring, summer and autumn), as well as the 25th and 75th percentile, in order to retain intra-seasonal variance. Remaining missing data in the Landsat time-series was imputed with a temporal moving window median (TMWM) approach. An accuracy assessment shows TMWM performs relatively better in Southern Europe and lower in mountainous regions such as the Scandinavian Mountains, the Alps, and the Pyrenees. We quantify the usability of the different component data sets for spatiotemporal machine learning tasks with a series of land cover classification experiments, which show that models utilizing the full feature space (30 m DTM, 30 m Landsat, 30 m and 10 m Sentinel-2) yield the highest land cover classification accuracy, with different data sets improving the results for different land cover classes. The data sets presented in the article are part of the EcoDataCube platform, which also hosts open vegetation, soil, and land use/land cover (LULC) maps created. All data sets are available under CC-BY license as Cloud-Optimized GeoTIFFs (ca. 12 TB in size) through SpatioTemporal Asset Catalog (STAC) and the EcoDataCube data portal.

Introduction

Over recent decades, the world has experienced rapid growth in earth observation (EO) technology. This has brought many benefits to various applied fields, however, it also brings new challenges to aspiring users: massive data volumes produced by EO sensors and in-situ monitoring networks require new specialized expertise and extensive computing capacity. Wagemann et al. (2021) list the following five key challenges to finding, accessing, and combining big environmental data: (1) limited processing capacity on user side, (2) growing data volumes, (3) non-standardized data formats and dissemination workflows, (4) too many data portals, and (5) difficult data discovery. Environmental data needs to be as accessible and useful as possible, while all its limitations, caveats and uncertainties need to be clearly documented to minimize the risk of error propagation. In addition, decision-makers require easy access to open environmental data and critical assessment tools in order to dynamically synthesize the information needed to address many critical environmental and economic challenges (Giuliani et al., 2017). The European Green Deal specifically (Sikora, 2021) requires a diversity of environmental information to reach its ambitious project goals, especially those focused on reaching climate neutrality, preservation of natural capital, modernization and simplification of the common agricultural policy (CAP), and connecting farms to forks, all whilst enabling the socio-economic transformation of rural and agricultural areas.

To enhance environmental data use for decision-making, several groups in different areas around the world have been putting effort in building EO data cubes: spatially aligned time-series of calibrated multi-dimensional observations (Giuliani et al., 2017); also see Liu et al. (2021), Lu, Appel & Pebesma (2018), Mirmazloumi et al. (2022). Some prominent examples of EO data cubes include the Earth System Data Cube (Mahecha et al., 2020), Digital Earth Australia (Lucas et al., 2019), Digital Earth Africa (Yuan et al., 2021), and the Swiss Data Cube (Chatenoux et al., 2021). Infrastructures such as the openEO Platform (https://openeo.cloud/), and Google Earth Engine (https://developers.google.com/earth-engine) can also be considered EO data cubes (Giuliani et al., 2020) due to the ease with which users can combine the various data sets hosted on these platforms.

Two important EO data sources in this context are Sentinel-2 and Landsat. Sentinel-2 has provided global coverage every 5 days since the launch of its second satellite (Sentinel-2B) in 2017, available freely from multiple sources such as https://scihub.copernicus.eu and https://earthexplorer.usgs.gov/. In recent years it has served as input data for various global and continental land cover mapping initiatives, such as ESA’s Worldcover (Van De Kerchove et al., 2021), Google’s DynamicWorld (Brown et al., 2022), and Sentinel-2 Global Land Cover (S2GLC) (Malinowski et al., 2020). The spatial resolution of Sentinel-2 sensors varies; the red, green, blue, and near-infrared (NIR) bands are available at 10 m resolution, while the two shortwave infrared (SWIR) bands are only available at 20 m.

The Landsat program is the world’s longest continuously running EO mission (Wulder et al., 2022). It is de facto the only option for assessing long-term dynamics as it provides an uninterrupted supply of satellite imagery since 1972. The entire archive was made available to the public in 2008, leading to widespread use, including refinement into data sets closer to analysis-ready status. The University of Maryland (UMD) Global Land Analysis and Discovery (GLAD) laboratory’s Landsat ARD product is another representative example of long-term EO data due to its free availability, global coverage, and its inclusion and harmonization of a succession of Landsat satellite sources (Potapov et al., 2020). The original data is available in 23 × 16-day scenes per year in scaled long format (Potapov et al., 2020). While this high temporal resolution and numerical precision provide a large amount of information for subsequent modeling and has been successfully utilized as such by teams with access to large computational resources (Hansen et al., 2022), the added benefit for land use land cover (LULC) classification compared with a compressed, more accessible form, has not yet been quantified. Furthermore, although the data set is nominally analysis-ready, we encountered the following limitations of using this data set for actual vegetation/land cover mapping with machine learning: The data volume of GLAD’s original archive (23 four-byte values, or 92 bytes per band per pixel per year) may exclude users without advanced computational capacity from performing country- or continental-scale analysis;

While multiple Sentinel-2 data sets are now available from 2015 (Copernicus Sentinel program), a harmonized, cloud-optimized product that is freely accessible regardless of institutional membership and computational resources could greatly increase global usage, especially among marginalized users.

While the aggregation to 23 16-day reflectance values increases coverage, gaps remain in the archive due to e.g., snow cover.

To maximize the usability of the produced data sets and facilitate future work in annual mapping, we have built a data cube and a data portal available on https://EcoDataCube.eu. It integrates various layers into a single seamless expandable and open access system. In this article we describe the key processing steps used to produce data cubes. We first explain the process of obtaining, gap-filling, artifact removal, and harmonization of EO images: gap-filled Landsat time series from 2000–2020, two Sentinel-2 time-series from 2018–2021 at varying temporal and spatial resolutions, and an optimized digital terrain model created with an ensemble machine learning approach. Finally, we provide examples of EcoDataCube usage and demonstrate case studies for which this data cube provided the main feature space, such as annual LULC maps (Witjes et al., 2022) and potential and realized tree distribution (Bonannella et al., 2022).

Methods and data

In this work we detail the processing and validation workflows of the following four data sets: 1. Quarterly spatiotemporal Landsat aggregates (median, 75th, and 25th percentile) of blue, green, red, NIR, SWIR1, SWIR2, and thermal bands at 30 m resolution between 2000 and 2020.

2. Two spatiotemporal aggregates (median, 75th and 25th percentiles) of Sentinel-2 between 2018 and 2021:

Annual blue, green, red, and NIR at 10 m resolution;

Quarterly blue, green, red, NIR, SWIR1 and SWIR2 bands at 30 m resolution;

3. An optimized digital terrain model (DTM) for Europe, created with an ensemble machine learning data fusion approach.

In order to quantify the extent to which these data sets complement each other as a single feature space for annual mapping with machine learning, we include a series of land cover classification experiments where we compare model performance when trained on different combinations of EcoDataCube layers. Figure 1 provides an overview of the general workflow and the resulting output and findings. These data sets all cover the exact same area, which is defined by all member and partner states of the European environment agency (EEA) in 2019, with the exception of Turkey (See Fig. 2A).

Figure 1 Overview of the general workflow with input, intermediate, and output data sets, as well as operations and evaluations of data set quality.

Figure 2 Overview of (A): the area of interest, (B): GLAD Landsat ARD tiles, (C and D): Sentinel-2 orbits and scenes used as input sources, and (E): the mosaicking algorithm that (1) computes quarterly composites, (2) mosaics the quarterly composites along orbital tracks, and (3) stitches the orbital track mosaics into a single data set.

Landsat

For this work, we used the landsat analysis-ready data (ARD) product developed by the UMD’s GLAD lab, a globally consistent analysis-ready data set for multi-decadal LULC monitoring (Potapov et al., 2020). It consists of 16-day time-series composites (23 per year, see Table 1) from Landsat 5, 7 and 8 which have been calibrated using MODIS surface reflectance.

Table 1 Overview of the start and end dates of the four temporal composite periods (quartiles) and of which GLAD interval IDs they were composed.

	Date	GLAD interval ID	
Quartile	Start	End	Start	End	
1	December 2nd of previous year	March 20nd	22 of previous year	5	
2	March 21st	June 24th	6	11	
3	June 25th	September 12th	12	16	
4	September 13th	December 1st	17	21	

Table 2 shows the bands, their spectral range in the different Landsat sources, and their spatial resolution. These time-series are freely available in 1-degree tiles (see Fig. 2B) and have been screened to flag pixels that likely contain clouds and their shadows in a quality assessment (QA) layer. While this data set is already a level 3 remote sensing product (i.e., temporal composites of gridded data), we aim to make it both more analysis-ready and easier to use by compressing it and imputing any missing values in a computationally efficient way that yields values suitable for classification tasks.

Table 2 Spectral bands used by Potapov et al. (2020) to create the GLAD Landsat ARD data set from multiple Landsat sensors.

Sensor	Landsat 5	Landsat 7	Landsat 8	
Time range	2000–2011	2000–2021	2013–2021	
Band	Wavelength (nm)	
Blue	450–520	441–514	452–512	
Green	520–600	519–601	533–590	
Red	630–690	631–692	636–673	
NIR	760–900	772–898	851–879	
SWIR1	1,550–1,750	1,547–1,749	1,566–1,651	
SWIR2	2,080–2,350	2,064–2,345	2,107–2,294	
Thermal	10,410–12,500	10,310–12,360	10,600–11,190	

Landsat temporal composites

In order to balance the trade-off between computation time of large areas while retaining as much temporal variability as possible, we aggregated the 23 annual GLAD Landsat ARD values into four annual quarterly period medians based on the astronomical seasons described by Trenberth (1983), which allows the four periods to act as a proxy for the four typical seasons in large parts of Europe: winter, spring, summer, and autumn. This allows us to match the beginning and end of each period with the 16-day intervals used by Potapov et al. (2020) (see Table 1 and Fig. 3B). We also calculated the 25th and 75th percentile of these aggregated values per pixel in order to maintain a measure of variance within each period that might be useful to recognize intra-annual dynamics. This yields 84 layers for each year (four quarterly periods × three percentiles × seven Landsat bands, see Fig. 3A) with varying amounts of no-data values based on cloud and snow cover.

Figure 3 Overview of the spatial and temporal resolution and covered time range of the presented Landsat and Sentinel-2 data.

(A) A comparison of the spatial and temporal resolution of the satellite imagery data sets included in EcoDataCube, as well as the available bands and time range covered per data set. Microsoft Bing Terrain screenshot © Microsoft Corporation. (B) The panel shows which GLAD ARD intervals were used to generate which quartiles, and how this compares to the two commonly defined seasons in Europe. GLAD ARD intervals that would be in different seasons depending on the definition are marked in bright orange.

Gap-filling

While the temporal composite aggregation reduces the number of no-data pixels in the time series at the cost of temporal resolution, gaps remain. While many gap-filling methods have been proposed, they are only well established for specific purposes (e.g., DINEOF (Alvera Azcarate et al., 2005) for ocean modeling, geostatistical neighborhood similar pixel interpolator (Zhu, Liu & Chen, 2012) for Landsat 7 Scan Line Corrector-off images), too computationally intensive to process multi-decade, continental-scale data (e.g., STAIR 2.0 (Luo et al., 2020) and linear temporal interpolation), and/or not available as maintained open source software.

In order to impute these remaining missing values in this multi-decade, continental-scale data set, we developed and implemented a custom gap-filling method: temporal moving window median (TMWM). The algorithm is designed to be computationally fast and suitable to gap-fill data for annual mapping for machine learning. It therefore only uses existing values in the data set instead of estimated values like averages or linear inter- and extrapolations, which makes sure that any imputed values are from the same feature space subsequent models are trained on. It fills gaps in a pixel by deriving median pixel values from its ‘temporal neighbours’. If the same pixel has a value for the same period in the next and/or previous year, TMWM takes the median of that period in the two ‘adjacent’ years. If the pixel had no value in the same period of the previous or next year, the ‘window’ expands to include values for that period in increasingly earlier and later years. If no value exists for the specified period in any year, TMWM will derive the pixel’s median value in the previous and next period of the same year. If that fails, the ‘window’ will again expand to include the previous and next period of increasingly earlier and later years. If no value can be found in these ways, the window encompasses all values in the entire time series of the pixel. If the pixel lacks data throughout the entire time series, the value is imputed with a local spatial average and assigned a QA value of 100.

TMWM attempts to derive a value for missing pixels in three phases. Within each phase, it will first try to use only eligible values within the last X and subsequent X years, where X is half the window size. If that fails, it will try to use values within double the search range. Then, it will try to use eligible values from any year in the time series. If that yields no result, it will move to the next phase, increasing the number of eligible values by including more intra-annual time periods in its search. These phases differ in which metric they calculate and which periods they use in the following way: The median of the same period from different years;

An average of the medians of the previous and next periods;

The median of all periods.

We validated TMWM’s performance on the temporal composite Landsat data from 2000 to 2020. This was done by sampling 100 pixels from each 6,750 30 km tiles and extracting the time series for the 50th percentile of each band and each period. If tiles were not completely covered by land, the number of sampled pixels was reduced in proportion. We then created a boolean mask based on whether the pixel had a value in that year. This ‘missing value mask’ was then inverted in order to introduce additional simulated gaps, reproducing existing patterns in which missing values occur in the data. We then used the TMWM method to fill the simulated gaps, and compare the imputed values to the original values. We quantified the performance of TMWM by deriving the root mean square error (RMSE) for each band and quartile. As different bands have different value ranges, we normalized the RMSE per band by dividing it by the range (maximum value minus minimum value) of that band’s values in the entire data set. This normalized RMSE (NRMSE) allows for a standardized comparison of performance across bands and years.

Sentinel-2

We created two Sentinel-2 2018–2021 time series of the study area: one series with annual values at 10 m resolution limited to the red, green, blue, and NIR bands, and one quartile/seasonal series at 30 m resolution which also includes the SWIR1 and SWIR2 bands. The data was processed in four steps (Fig. 2E): Computing temporal composites (annual and quarterly) for each Sentinel-2 tile;

Reprojecting and resampling the tiled composites to EPSG 3035 (https://epsg.io/3035) at 10 and 30 m resolution;

Mosaicking the resampled composites over their respective orbital tracks;

Stitching the orbital mosaics together.

The Sentinel-2 mosaics were built from Sentinel-2 Level 2A (S2L2A) imagery (BOA reflectance generated with scene classification and atmospheric correction algorithms), for six bands (see Table 3). The mosaics span 1,028 tiles. Each scene over these tiles, imaged over the time period from winter 2017/2018 (2.12.2017) to winter 2020/2021 (1.12.2021), was collected from the AWS Sentinel-2 repository, which is hosted as a Requester Pays S3 bucket (accessible without charge from AWS instances). The mosaicking and temporal aggregation was performed with the s2mosaic functionality of the eumap python package1.

Table 3 Sentinel-2 bands included in the data cube with their respective wavelengths and original imaging resolutions.

Band number	Band name	Wavelength (nm)	Resolution (m)	
B02	Blue	496.6–492.1	10 m	
B03	Green	560–559	10 m	
B04	Red	664.5–665	10 m	
B08	NIR	835.1–833	10 m	
B11	SWIR1	1,613.7–1,610.4	20 m	
B12	SWIR2	2,202.4–2,185.7	20 m	

Sentinel temporal composites

We created annual composites of the four 10 m resolution Sentinel-2 bands: green, blue, red, and NIR. We aggregated all scenes captured between March 21st and December 1st to three percentiles: 25th, 50th (median) and 75th, excluding all pixels flagged in the S2L2A cloud and cloud shadow masks. These were then resampled to EPSG:3035. We also created quarterly percentile composites of all bands (green, blue, red, NIR, SWIR1, and SWIR2) with date ranges matching the Landsat quartile periods. The tile-wise composites were resampled to 30 m resolution in EPSG:3035 and aggregated to the same quartiles as the Landsat ARD data. Since scenes that are acquired along the same orbital track are very likely to be imaged in equivalent conditions, seamless mosaicking is possible by averaging the overlapping pixels. A total of 32 orbital tracks were used (Fig. 2D). The final mosaicking was done by stitching together the orbital mosaics with weighted averaging of overlapping pixels. For each pair of overlapping pixels, the relative distance from their respective orbital track was calculated (from 0 to 1, with 1 being the distance to the neighboring track). Overlapping pixels with a relative distance within the range of 0.4 to 0.6 (inclusive) were averaged by using their relative distances as weights, while pixels with a relative distance below 0.4 were designated the correct value, regardless of overlap, as artifacts were often observed at relative distances above 0.6.

Digital terrain model

Although a continental-scale DTM called “EU-DEM” already exists, it is based on SRTM and ASTER GDEM (Józsa, Fábián & Kovács, 2014); we have built a DTM for the study area using more detailed and more up-to-date elevation products: MERIT DEM (Yamazaki et al., 2019), ALOS AW3D (Takaku et al., 2018) and GLO-30 (https://doi.org/10.5270/ESA-c5d3d65). To generate the best estimate of the land surface/terrain elevation, we used 10 input variables, obtained by overlaying the training points on five elevation and five auxiliary raster data sets (Table 4), and an ensemble machine learning approach trained on a random sample of seven million (randomly sampled from the 28 million points available) global ecosystem dynamics investigation (GEDI) points and 2 million ICESat-2 points. We specifically used GEDI level 2B points elev_lowestmode column and ICESat-2 (ATL08) h_te_mean column, which in both cases represent the lowest elevation observed i.e., the most likely bare ground height. We combined these two ground-truth data sets in a total of nine million data points, and then built a machine learning model that we used to predict height without canopy and report predictions errors.

Table 4 Overview of data sets used as input for the ensemble that produced our DTM.

Dataset	Producer	Source	
MERIT DEM	University of Tokyo Global Hydrodynamics lab	Yamazaki et al. (2017)	
EU-DEM	European environmental agency (EEA)	Mouratidis & Ampatzidis (2019)	
ALOS AW3D0	JAXA earth observation research center (EORC)	Tadono et al. (2014)	
GLO-30	European space agency (ESA)	European Space Agency (ESA) (2018)	
Canopy height	UMD GLAD	Potapov et al. (2021)	
Surface water probability	European commission joint research centre (JRC)	Pekel et al. (2016)	
Tree cover	Global forest watch	Hansen et al. (2013)	
Bare ground cover	UMD GLAD	Hansen et al. (2013)	
Pan-European land cover	Humboldt University of Berlin	Pflugmacher et al. (2019)	

The ensemble model was composed of a random forest, a cubist model, and a generalized linear model in the mlr R package. These models make separate predictions using all input variables. Their estimates are then used as input for the makeStackedLearner meta-learner function (Bischl et al., 2016); in this case a linear regression model that makes the final elevation estimate for each pixel. This approach can be compared to the approach of Hawker et al. (2022) who produced a global map of elevation with forests and buildings removed, however, in our approach we also use the continental EU-DEM data set and numerous additional layers, at the cost of higher complexity and computational effort. We also validated the four DEMs used as input for the ensemble by comparing their values with the GEDI/ICESat-2 training data set, which can be considered ground-truth data.

Note that the predicted elevations are based on the GEDI data, hence the reference water surface (WGS84 ellipsoid) is about 43 m higher than the seawater surface for a specific EU country. Before modeling, we corrected the reference elevations to the Earth Gravitational Model 2008 (EGM2008) by using the 5-arcdegree resolution correction surface (Pavlis et al., 2012).

We assessed the accuracy of the ensemble model and the resulting DTM by performing k-fold spatial cross-validation (Lovelace, Nowosad & Muenchow, 2019). We divide the study area into square blocks of 30 × 30 km, which are grouped to provide folds for the cross-validation approach. Because the publications describing the input DSMs do not report accuracy with comparable metrics or criteria, we also validated each input data set with our training/cross-validation data set i.e., by using the GEDI and ICESat-2 points. This provides an objective comparison between each input DTM, and the cross-validation predictions of the ensemble. The produced Ensemble DTM of Europe and prediction errors were provided as GeoTIFFs using Integer format (elevations rounded to 1 dm) and have been converted to Cloud Optimized GeoTIFFs using GDAL 3.1.4.

Land cover classification experiments

Because the intended purpose of the presented data sets is to facilitate annual mapping with machine learning, we compare the usefulness of 30 m Landsat, 30 m Sentinel-2, and 10 m Sentinel-2 for land cover classification. We do this by training several random forest (RF) models on 300,543 observations from the European land use and land cover survey (LUCAS) data set harmonized by d’Andrimont et al. (2020) to predict the 8 LUCAS level-1 land cover classes; each of these RF models uses a different combination of the feature space provided by the data cube (Sentinel-2 10 and 30 m, Landsat, and DTM). Additionally, to investigate the added value of the multi-decade harmonized Landsat time series, we trained RF models on all 1.4 million available LUCAS observations from 2000–2020. For each classification task we used a RF classifier with 100 trees, one minimum sample per leaf, and two minimum samples per node, implemented in Python 3.8.6 using Scikit-Learn. The maximum number of features per tree was set to the square root of the amount of total features. We assess the performance of each model through both five-fold cross-validation on its training set, and by validating each model on one randomly sampled left-out test data set of 33,394 LUCAS observations from 2018 and 2019. The different combinations of time range, data set usage, and train/test points are presented in Table 5.

Table 5 Overview of land cover classification experiments that were performed to quantify the added value of each data set to the data cube.

Satellite	Resolution	Time range	DTM used	Training points	
Landsat	30 m	2000–2020	Yes	1,443,227	
Landsat	30 m	2000–2020	No	1,443,227	
Landsat	30 m	2018–2021	Yes	300,543	
Landsat	30 m	2018–2021	No	300,543	
Sentinel-2	30 m	2018–2021	Yes	300,543	
Sentinel-2	30 m	2018–2021	No	300,543	
Sentinel-2	10 m	2018–2021	Yes	300,543	
Sentinel-2	10 m	2018–2021	No	300,543	
Sentinel-2	10 m + 30 m	2018–2021	Yes	300,543	
Sentinel-2	10 m + 30 m	2018–2021	No	300,543	
Landsat + Sentinel-2	30 m	2018–2021	Yes	300,543	
Landsat + Sentinel-2	30 m	2018–2021	No	300,543	
Landsat + Sentinel-2	10 m + 30 m	2018–2021	Yes	300,543	
Landsat + Sentinel-2	10 m + 30 m	2018–2021	No	300,543	

Results

Landsat

Downloading the 2000–2020 time-series of 16-day composites for 1,149 1-degree GLAD geotiffs of 72.3 GB each amounted to approximately 81 TB of data, or 1 TB per band and year. Compressing the scaled long integer values to byte format and aggregation into temporal composites reduced this to 29 GB per band per year, constituting a size reduction of about 97.1%. Removing all pixels not labeled as having “clear sky” in the GLAD metadata, resulting in an average of 5.83% empty pixels in spring, 19.7% in summer, 11.73% in autumn, and 54.29% in winter, when aggregated to quarterly temporal resolution. Figure 4 shows that, on average across all years, more gaps occur in Scandinavia and the northern Baltic countries. Figure 4E shows that the winter quartile of each year consistently had the highest number of gaps, followed by summer in all years except 2003. It also shows a clear reduction in gaps after the winter of 2012–2013 and the inclusion of Landsat-8 data in the archive.

Figure 4 Percentage of gaps per pixel in the Landsat 30 m data between 2000–2020.

(A–D) The annual average, calculated per 30 km tile (one million pixels) for each of the four quartiles that the GLAD Landsat ARD product was aggregated to. (E) The percentage per year and quartile.

Sampling 100 gap-filling validation pixels for each 30 km tile in proportion to its land area resulted in 566,454 pixels from 6,750 tiles. Table 6 shows the average gap-filling NRMSE respectively per band and quartile. Our validation shows that the lowest gap-filling performance was on the NIR band in Q2 (Spring) with a NRMSE of 4.41%, while the thermal band in Q2 was filled the most accurately with a NRMSE of 0.66%. More generally, Q3 (Summer) was gap-filled the most accurately with an average NRMSE of 2.09%, while Q4 (Winter) was gap-filled the least accurately with an average NRMSE of 2.36%. Figure 5 shows the spatial variability of gap-filling NRMSE per quartile, indicating a consistent higher error rate in mountainous areas, especially the Alps and the Scandinavian mountains.

Table 6 Gap-filling validation NRMSE (in percentage) per band and quartile, as well as band & quartile averages.

Band	Q1 (Winter)	Q2 (Spring)	Q3 (Summer)	Q4 (Autumn)	Average	
Blue	2.10	1.79	1.84	2.15	1.97	
Green	1.64	2.03	1.65	1.99	1.83	
Red	2.37	2.25	2.30	2.03	2.24	
NIR	3.74	4.41	3.69	4.34	4.05	
SWIR1	2.46	2.88	2.43	2.92	2.67	
SWIR2	2.20	2.18	2.08	2.37	2.21	
Thermal	0.68	0.66	0.67	0.71	0.68	
Average	2.17	2.31	2.09	2.36	2.23	

Figure 5 Map of average gap-filling NRMSE per 30 km tile in the study area.

Results show consistently higher values in mountainous regions, and lower values in Southern Europe.

Sentinel-2

A total of 190,884 Sentinel-2 scenes were processed, ranging from 13 to 119 scenes per tile and quartile (see Table 7), amounting to a data set size of roughly 15.5 TB per quartile, or 62 TB per year. The input data for each annual composite of Sentinel-2 10 m RGB+NIR was an average of 62 TB per year. Aggregating them to annual composites reduced the size to roughly 0.3 TB per band, or 1.2 TB for the four bands, resulting in 98.1% compression. The 10 m resolution product had an average of 0.397% gaps per year, with the median among 30 km tiles being 0%. Figure 6 shows that most 30 km tiles with gap percentages above 0.1% occur on tiles next to water.

Table 7 Number of Sentinel-2 scenes processed per Sentinel-2 tile and quarter.

Quartile	Max	Min	Mean	Standard deviation	
Q1 (Winter)	84	16	47.95	11.82	
Q2 (Spring)	119	19	53.07	17.49	
Q3 (Summer)	100	16	43.87	14.65	
Q4 (Autumn)	81	13	40.07	10.38	

Figure 6 Overview of gaps per pixel in the Sentinel-2 time series.

Average annual percentage of gaps per quartile and 30 km tile (A–D) and in total (E) for the 30 m data, as well as average annual percentage per 30 km tile for the 10 m data (F).

The input data for each annual set of Sentinel-2 quarterly composites at 30 m RGB+NIR+SWIR were identical to those used for the annual 10 m composites, amounting to an average of 62 TB per year. Aggregation to quarterly composites reduced the size to 0.2 TB per quarter, or 0.8 TB yearly, resulting in a compression of 98.7%.

Figure 6E shows that most gaps occur each year in the first quarter, especially in 2018. Except in 2018, the second highest number of gaps occurred in quarter 4 (September–December). Figures 6A–6D shows that across all years, Northern Scandinavia has the least gaps in quarter 2 (March–June), and that relatively more gaps occurred in a stripe pattern across Europe in quarter 1 (December–March).

Digital terrain model

Results of modeling terrain with five million GEDI and two million ICESat-2 elevation measurements, using four existing DEMs and six auxiliary data sets as input variables, show a maximum cross-validation RMSE of 6.54 m for absolute accuracy of predicting terrain (bare-earth) height with majority of errors between 2–3 m (25% and 75% quartiles). MERIT DEM (Yamazaki et al., 2019) is the most correlated DEM with GEDI and ICESat-2 points, most likely because it has been systematically post-processed and the majority of canopy problems have been removed (Fig. 7).

Figure 7 Comparison of ensemble DTM (left) and the AW3D digital surface model for the Pannonian plane in Eastern Croatia (right).

Tree height visible on AW3D seems to be systematically removed with the help of machine learning. Grid showing 30 km tiles.

Figure 8 shows the average RMSE of the four input DEMs per 30 km tile and the standard deviation of these four values. Our results suggest that MERIT and EUDEM are less accurate in large parts of Sweden, while GLO30 and AW3D are less accurate in central Europe. Furthermore, red lines are visible on these maps that match GEDI orbits and no natural features. Table 8 compares the RMSE of each DEM and the Ensemble DTM. This shows that MERIT DEM had both the lowest RMSE (8.45 m) and the highest variable importance in the model. In summary, our results show that the DTM produced by the ensemble is approximately 2 m more accurate than MERIT in the area of interest. A copy of the all inputs, regression-matrix and outputs produced, including the code used to fit models and produce predictions, is available via https://doi.org/10.5281/zenodo.4056634.

Figure 8 Results of comparing GEDI and ICESat-2 measurements to AW3D, GLO30 EUDEM and MERIT values in 30 km tiles covering the study area.

(A) The average RMSE across the four data sets; (B) the standard deviation among RMSE values, representing the disagreement between the four data sets. Note the straight lines of higher average RMSE values in e.g., Iberia, which (B) suggests are consistent across all four data sets. Instead, the data sets disagree most in parts of Northern Europe, especially Iceland.

Table 8 RMSE of the four input DEMs and the output DTM produced with ensemble machine learning. The RMSE of the input DEMs was acquired by comparing them to the values of 7 million GEDI/ICESat-2 points. The RMSE of the DTM was acquired through spatial cross-validation of the ensemble model. Variable importance from the random forest in the ensemble is included for the input DEMs.

Dataset	RMSE	Variable importance	
MERIT	8.451	430 B	
AW3D	9.858	291 B	
EUDEM	9.806	132 B	
GLO30	9.900	201 B	
EcoDataCube DTM	6.544	NA	

Land cover classification tests

The random forest utilizing all four data sets (DTM, Landsat, Sentinel-2 10 and 30 m) achieved the highest cross-validation scores (0.761) and second-highest test score (0.767). In general, models with multiple data sets in their feature space (DTM, Landsat, Sentinel-2 at 10 m spatial resolution, and/or Sentinel-2 at 30 m) outperformed models using a single data set. However, the highest test score of 0.774 was achieved by the Landsat model trained on 300,543 LUCAS observations spread out across 2000–2020, while this model had a relatively low cross-validation score of 0.715 (see Table 9). It appears that models utilizing DTM variables achieved higher cross-validation and test F1-scores in every experiment except for the Sentinel-2 10 m model. This model also achieved the lowest cross-validation and test scores.

Table 9 Overview of land cover classification tasks that were performed to compare the usefulness of the different products, with weighted averaged F1-scores from cross-validation and from predicting on test set.

The highest scores are shown in bold.

Dataset	Resolution	Time range	DTM	Training points	CV F1-score	Test F1-score	
Landsat	30 m	2000–2020	Yes	1,273,518	0.710	0.738	
Landsat	30 m	2000–2020	No	1,273,518	0.698	0.730	
Landsat	30 m	2000–2020	Yes	300,543	0.715	0.774	
Landsat	30 m	2000–2020	No	300,543	0.707	0.768	
Landsat	30 m	2018–2021	Yes	300,543	0.731	0.742	
Landsat	30 m	2018–2021	No	300,543	0.724	0.736	
Sentinel-2	30 m	2018–2021	Yes	300,543	0.746	0.753	
Sentinel-2	30 m	2018–2021	No	300,543	0.741	0.748	
Sentinel-2	10 m	2018–2021	Yes	300,543	0.705	0.713	
Sentinel-2	10 m	2018–2021	No	300,543	0.706	0.715	
Sentinel-2	10 m + 30 m	2018–2021	Yes	300,543	0.758	0.760	
Sentinel-2	10 m + 30 m	2018–2021	No	300,543	0.751	0.756	
Landsat + Sentinel-2	30 m	2018–2021	Yes	300,543	0.750	0.757	
Landsat + Sentinel-2	30 m	2018–2021	No	300,543	0.747	0.755	
Landsat + Sentinel-2	10 m + 30 m	2018–2021	Yes	300,543	0.761	0.767	
Landsat + Sentinel-2	10 m + 30 m	2018–2021	No	300,543	0.758	0.765	

Figure 9 shows that models trained on data sets with multiple satellite sources and resolutions generally outperformed single-source or single-resolution models. Figure 9A shows that “Shrubs” and “Wetlands” were more accurately classified by models only using 30 m data sets. Figure 9B shows that Sentinel-2 data was more useful for classifying “Artificial” and “Water areas”, while being less useful for classifying “Shrubs” and “Wetlands”. For these classes, Landsat data yielded higher F1-scores. Using the DTM and its derived variables generally yielded slight performance increases, except for “Crops” and “Bare Ground”. Figure 9A shows that including the DTM in the feature space mainly lead to higher accuracy when classifying “Shrubs” and “Wetlands”.

Figure 9 Test F1-score of random forests trained to classify eight LUCAS land cover classes, shown per class and aggregated based on which data sets were included in their feature space ((A) spatial resolution, (B) satellite source, and (C) DTM).

(A) Graph showing that models trained on both 10 m and 30 m resolution data generally achieved the highest classification accuracy, but that models trained solely on 10 m (which is only Sentinel-2) classified water more accurately than models trained either 30 m data sets or a combination of 30 m and 10 m. In particular, the “Shrubs”, “Grass”, and “Wetlands” classes were predicted much less accurately by the 10 m models. Combining 10 m and 30 m resolution data lead to the largest performance increase for the “Artificial” class. (B) Graph showing that models trained on data from both satellite types outperformed other models on every class except “Water”. (C) Graph showing that models including DTM data in their feature achieved higher accuracy when classifying “Shrubs” and “Wetlands”.

Discussion

We have demonstrated a full methodological framework for processing various EO data for the purpose of producing an open Data Cube for Europe, evaluated all steps, and investigated the value of combining its component data sets using machine learning applications. Our key findings indicate that: Combining all four data sets produced in this work (DTM, Landsat 30 m, Sentinel-2 30 m and Sentinel-2 10 m) yields the highest land cover classification accuracy, with different data sets improving the results for different land cover classes;

When used separately, the 2000–2020 Landsat data set can be used to model longer time series. In our experiments, models trained on LUCAS observations in this longer time span generalized better than those trained on an equal amount of points, but only sampled from 2018–2019;

Ensemble machine learning can be used as a data fusion technique to combine global elevation models and create an optimized DTM that is more accurate in the area of interest, based on an independent validation;

Accuracy and visual assessment of the four input DEMs suggests that DTM could still be much improved if countries would donate their national higher resolution elevation data.

In the next sections we discuss some remaining limitations of the Data Cube and suggest possible strategies to overcome them.

Suitability of temporal composite design

We recognize that the choice for aggregating the 23 × 16–day GLAD measurements into temporal composites that approximate the typical four seasons in central Europe imposes some limitations. Firstly, seasonality differs per region, even within the study area. This can cause differences in performance between regions with a matching seasonality and regions with a different one. This may be a potential explanation for the poorer accuracy of land use/land cover classification along the Mediterranean coast in Witjes et al. (2022), for which this data set provided a substantial part of the feature space. Secondly, the loss of temporal resolution likely hinders the accurate classification of dynamic classes that are distinguished by intra-seasonal variation, such as different crop types (Vuolo et al., 2018) or other modeling tasks involving vegetation phenology (Zhao et al., 2011). However, Zhao et al. (2022) found that choosing an appropriate temporal compositing strategy can reduce the need for a higher temporal resolution.

While a monthly aggregation would likely help solve these issues, but would also pose new challenges: it would be more complicated to derive from the 23 measurements, as they do not perfectly match the 12 months in the Gregorian calendar. In addition, a monthly aggregation would retain more gaps in the data. While the quarterly aggregation has less temporal resolution, the three percentiles quantify some intra-seasonal variability when multiple pixel values are available per quarter. This approach retains information on the variability in the growing seasons, while reducing the lack of data in an efficient way.

Another solution would be to use a non-symmetrical approach, where the growing season is divided into smaller time units, while the non-growing season is aggregated to a higher extent. As there are far fewer gaps in the quartiles roughly covering the growing season (Q2 and Q3, March–September; see Fig. 4), this approach would yield a data set that is: (1) more detailed where it matters, (2) requires less gap-filling, and (3) might be more suitable for gap-filling with TMWM. Such a technique, however, should only be used when constructing data cubes of areas with homogenous seasonal dynamics. Further research into the optimal temporal aggregation method for different subsequent modeling tasks would likely improve the usefulness of resulting data sets.

Gains and limitations of gap-filling with TMWM

The TMWM algorithm is computationally efficient and only imputes sets of existing combinations of pixel values across bands. It does have important drawbacks, however: firstly, in regions where data for a specific period is extremely sparse or non-existent (e.g., Northern Scandinavia in winter), data for this period will be almost completely derived from other periods. This can severely hamper the performance of classification tasks when a model needs intra-annual dynamics to distinguish certain classes. For example, our validation suggests that the Landsat RGB bands are easier to gap-fill with TMWM than infrared bands, especially NIR. This phenomenon is more pronounced in spring, which may be caused by the more dynamic and variable nature of vegetation in that period each year. Filling NIR data with values from other seasons may be exceptionally problematic in this respect.

Secondly, any model trained on data using this method may be less suitable for the timely detection of changes between predicted classes. Because TMWM prioritizes previous and subsequent years when imputing missing values, the resulting feature space may stay constant while the actual situation on the ground has changed. This may make the chosen combination of temporal aggregation and gap-filling less suitable for annual change analysis.

The qa_f layers included for every year and quartile at https://stac.ecodatacube.eu allow users to programmatically identify all gap-filled pixels and replace them with a gap-filling method that is optimal for their own subsequent modeling.

Finally, no gap-filling was implemented on the Sentinel-2 data sets. Although the total percentage of gaps is very low in most areas (see Fig. 6), the Sentinel-2 data is not fully complete and therefore not 100% analysis-ready.

The validation method used to assess the accuracy of the TMWM algorithm was chosen because it reproduces existing nodata patterns as observed in the intra and inter-annual dynamics of our data set, which we expect to yield to a more realistic evaluation of its performance on real-world data. However, mirroring the time-series occurrence of gaps as a validation mask simulates a larger number of missing values than their actual occurrence. Because the algorithm subsequently has less data to derive filling values with, this may lead to an underestimation of the actual accuracy.

Applicability and limitations of digital terrain model

We validated each of the four input DTMs on the harmonized GEDI/ICESat points that we used to train our model, allowing a comparison between these data sets and with the predictions made by our ensemble. It must be noted that a thorough accuracy assessment of GEDI and ICESat-2 for generating 30 m resolution DTMS on an European scale is not available. This means that it is hard to quantify the highest attainable accuracy when using it as training data. We did notice, however, possible artifacts in the GEDI data. Figure 8 shows that enough GEDI points matched poorly with each input DEM in line-shaped groups for them to cause relatively large average RMSE values in the 30 km spatial tiles. The effect is especially noticeable in Iberia, and does not match any natural features. It does, however, match the GEDI orbit track. This suggests that there might be some orbit issues affecting the local accuracy of the predictions.

For the ensemble DTM we have produced, we noticed that in many places canopy height is still visible on the hillshading images, indicating that even after using the canopy height, the true terrain elevation in forests is overestimated. Additional filtering is needed to remove human built objects in urban areas. Our Ensemble DTM has not been hydrologically corrected and will require additional processing before it can be used for spatial modeling. Furthermore, several EU countries such as Belgium, the Netherlands, and Denmark have high-resolution terrain models built from LIDAR surveys. The ensemble DTM could be further improved by merging such publicly available data. Most importantly, it should be compared with the recently published 30 m global map of elevation with forests and buildings removed (Hawker et al., 2022).

Usefulness of EO data for land cover and land use mapping

The land cover experiments, while limited in scope, clearly show that combining different data sets in the data cube improved modeling performance for classification tasks. This demonstrates the value of spatially, temporally, and spectrally harmonized multivariate data cubes such as the one presented in this work. For some of the eight land cover classes, the combined models were outperformed by a model using only a specific data set or resolution (e.g., Sentinel-2 models when classifying water areas, and 30 m resolution models when classifying wetlands).

The different Landsat-only experiments also suggest that sacrificing spatial resolution in order to access longer time-series of training data may yield better subsequent modeling results; the 300 K 2000–2021 Landsat model outperformed the other Landsat models on the test data set. This matches findings by Witjes et al. (2022) and Pflugmacher et al. (2019). However, the lower performance of the 1,273 K 2000–2020 Landsat model suggests that using a larger training data set does not necessarily improve model performance.

Shrubland and bare land were consistently the least accurately classified land cover classes in each experiment. This was particularly the case for models trained only on 10 m Sentinel-2 data (see Fig. 9A), even when taking into account their lower performance across all classes. This lower performance was likely affected by the lower number of available bands, i.e., the lack of SWIR and NIR. This possible explanation is supported by the fact that models combining 30 and 10 m Sentinel-2 data outperformed Landsat-only models on bare land, The model trained only on Sentinel-2 data however did outperform other models when classifying water areas, achieving a slightly higher score than even the model combining all data sources. These findings, combined with indications that the Sentinel-2 SWIR bands are highly useful for distinguishing between different tree species (Immitzer et al., 2019) suggest that incorporating both 10 and 30 m Sentinel-2 data sets as part of the data cube enhances subsequent land cover modeling efforts.

The cost of accessibility and analysis-readiness

The ultimate goal of this work is to present an analysis-ready data cube that is as useful as possible to as many different users as possible, with an emphasis on time-series classification tasks. To this end, we have greatly—sometimes by more than 90%—reduced the size of the input data, reduced the number of gaps to below 1% for both Landsat and Sentinel-2 time-series, and made all products freely accessible through modern technologies and formats such as COG and SpatioTemporal Asset Catalogs (STAC) without requiring any form of user registration. However, this emphasis on accessibility and analysis-readiness comes at the cost of both temporal and spectral detail.

We aggregated a 23-part Landsat time-series and a Sentinel-2 time-series with a highly variable number of observations (13–119) into four quartiles with three percentiles each: a standardized total of 12 values. While this low temporal resolution approach requires less gap-filling and makes the data set less susceptible to error propagation, it may not be detailed enough to detect certain classes in some classification tasks such as those relying on vegetation phenology. Our land cover experiments did not try to distinguish between different vegetation types or inform vegetation growth models, which might only be discernible when having a higher number of observations in the growing season (Zhao et al., 2011). On the other hand, Vuolo et al. (2018) present significant increases in observer’s accuracy (OA) of crop type classification when using multi-date data, and suggest that this method alleviates the issue of finding the optimal temporal window where the highest single-date accuracy can be achieved.

We did not analyze the effects of compressing the Landsat values from long unsigned integer (0–40,000) and unsigned 16-bit integer (0–65,535), respectively, to Byte (0–255). This loss in precision may lead to similar limitations, especially in the case of infrared for vegetation-related modeling. However, Bonannella et al. (2022) achieved high accuracy (0.81–0.89) using the Byte-scale Landsat data when classifying multiple different tree species, suggesting this may not, after all, be a significant limitation.

Future work

While the EcoDataCube layers presented in this work bridge a large part of the gap between data and users through a consistent design philosophy, they are not completely finished. Notably, gaps remain in both Landsat and Sentinel-2 data. Work continues on a more efficient and effective gap-filling methodology optimized for classification tasks with probability-based post-processing such as the methodology used in Witjes et al. (2022). The effects of different temporal resolutions and percentile usage will need to be compared in order to reach the optimal method for any specific task. Purely relying on time-series of the same pixel may be too limiting for areas with frequent and consistent gaps, such as Scandinavia and mountainous regions. A less stringent QA-informed removal of pixels could reduce this issue, but might require additional processing steps. Furthermore, GLAD will discontinue its current ARD Landsat archive, publishing a recomputed version that will be continued in the foreseeable future. While this necessitates a recomputation of all current EcoDataCube Landsat layers as well, it provides an excellent opportunity to compare and implement the next generation of temporal composition and gap-filling techniques.

The EcoDataCube platform already hosts more data sets that follow the same design philosophy of analysis-readiness and accessibility (Wagemann et al., 2021), allowing rapid and user-friendly comparisons and synthesis (see Fig. 10). Examples are the potential and realized distribution of 16 tree species (Bonannella et al., 2022), monthly airborne fine particulate matter levels (Ibrahim et al., 2022), 43 CORINE land cover classes (Witjes et al., 2022), and daily aerosol optical depth levels (Ibrahim et al., 2021).

Figure 10 Screenshot of the EcoDataCube.eu data viewer showing (A): the probability of pasture land cover in 2019 near Copenhagen, Denmark, and (B): Landsat-derived NDVI in the summer of 2019.

We aim to continuously extend the feature space of the data cube, both by producing new data sets and hosting harmonized products created by third parties. For instance, a backwards estimation of Sentinel-2 values to cover the same time period as the Landsat data set is under consideration. Data fusion approach such as the FORCE (Frantz, 2019) provides a systematic solution for this and will likely be used for this purpose. There are thousands of data sets that could potentially be integrated in, and shared on, the EcoDataCube. For instance, with proper documentation and spatiotemporal harmonization, many of the over 1,500 data sets from the European Environmental Agency’s archive (https://www.eea.europa.eu/data-and-maps) could be added to the feature space and breadth of analyses offered on the platform.

Conclusions

With the EcoDataCube data sets and platform, we present a spatiotemporally consistent, transparently and reproducibly processed continental-scale data set that is hopefully as accessible as possible across platforms. We intend it to reach the widest possible user base and to be put to as many different uses as possible—generating more value for society—while also facilitating collaboration and reproducibility. The intended uses of EcoDataCube include vegetation, soil, land cover and land use mapping projects, environmental monitoring by the EEA, and automated generation of data for statistical offices such as Eurostat (https://ec.europa.eu/eurostat).

The EcoDataCube data sets are hosted through modern cloud-based solutions that are both humanly and programmatically accessible without limitation and with minimal effort through STAC and the EcoDataCube platform, and will be continuously updated, maintained, and expanded.

The spatiotemporal harmonization and gap-filling processes described in this work ensure that users can easily combine different data sets to perform their analyses without the need for extensive preprocessing. The included validations and published QA data sets accompanying the TMWM-gap-filled Landsat data ensure that all limitations of the data set can be easily analyzed and communicated. As we continue our work to create and host more open spatiotemporal analysis-ready data, we encourage and invite all interested parties to use these data sets and to provide feedback, especially on inaccuracies or limitations, so that these can be addressed in future versions.

The back-end and front-end solutions for EcoDataCube.eu were co-developed jointly with GiLAB Ltd, Belgrade, Serbia (https://gilab.rs/).

Additional Information and Declarations

Competing Interests

Author Contributions

Data Availability

1 Code is available at https://gitlab.com/geoharmonizer_inea/eumap/-/tree/master/eumap/datasets/eo/s2mosaic, documentation at https://eumap.readthedocs.io/en/latest/notebooks/09_sentinel2_mosaicking.html.

Josip Križan & Luka Antonić are employed by MultiOne. The authors declare that they have no competing interests.

Martijn Witjes conceived and designed the experiments, performed the experiments, analyzed the data, prepared figures and/or tables, authored or reviewed drafts of the article, and approved the final draft.

Leandro Parente conceived and designed the experiments, performed the experiments, authored or reviewed drafts of the article, and approved the final draft.

Josip Križan conceived and designed the experiments, performed the experiments, authored or reviewed drafts of the article, and approved the final draft.

Tomislav Hengl conceived and designed the experiments, performed the experiments, prepared figures and/or tables, authored or reviewed drafts of the article, and approved the final draft.

Luka Antonić conceived and designed the experiments, performed the experiments, analyzed the data, prepared figures and/or tables, authored or reviewed drafts of the article, and approved the final draft.

The following information was supplied regarding data availability:

The Python code used to download the original Sentinel-2 data and GLAD Landsat archives and aggregate them into temporal composites is available at GitHub: https://gitlab.com/geoharmonizer_inea/eumap under MIT license.

The code used to perform the landcover classification experiments can be found on https://gitlab.com/research_m_witjes/ecodatacube. The codebase primarily uses components of the “eumap” python package, which is available from https://eumap.readthedocs.io/en/latest/.

The accompanying Docker image can run all python code used for this article. Data from all Landsat and Sentinel bands, quarters, and percentiles is available under a CC-BY license and can be downloaded through STAC at https://stac.ecodatacube.eu. It can also be explored and accessed through the EcoDataCube platform. All data sets are presented in the ETRS89-extended/LAEA Europe projection (EPSG:3035).

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
