# Peer review of "Ecodatacube.eu: analysis-ready open environmental data cube for Europe"

_PeerJ, doi:10.7717/peerj.15478_

## Round 0.1 · original submission · Major Revisions

Please revise this manuscript based on the reviewer comments.

Reviewer 1 ·

Basic reporting

The article aims to present a novel data cube for Europe, mainly for land cover / land use change detection. However, there are several elements which need major revision:

(1) in general, I recommend to have the manuscript be checked by a fluent English speaker as essential linguistic improvements can be made

(2) the manuscript at this state lacks a clear structure and from the introduction I cannot identify the aim and objective of the manuscript. The authors seem to aim to introduce the data cube, but it is not clear how the developed data cube is a valid addition and what specific gap it fills. Since the objective of the manuscript is not clear in the introduction, the other parts of the manuscript fail to respond to the hypotheses.

(3) There is a sheer amount of figures and tables, some of them are never referenced in the text. As a reader I lost oversight of the sheer amount of information that is presented to me. I recommend to make a selection of relevant figures and tables, after the objective of the manuscript has been defined more in detail.

(4) Literature referenced in Introduction and Discussion is sometimes not sound. E.g. some initiatives are listed as data cubes, which are initiatives or a company, but not a data cube solution.

Experimental design

As stated above, the manuscript tries to cover too many aspects and results and this results in a lack of clarity and structure. The data cube and data preparation is presented in detail, but then also a novel Temporal Moving Window Median approach is presented. After the data cube is created, there is a section of 'land cover experiments' whose objective I do not understand and if important, the section is not well enough explained and documented.

The identified knowledge gap and how the developed data cube fills this gap does not become apparent in the introduction and needs to be better elaborated throughout the manuscript.

For example, it can be valid to only focus on the data cube itself, how it was created and the data it contains. Additionally, it can be of value to feature how 'users' can benefit from the cube, e.g. how the cube can be accessed and what kind of analyses users can do with it (as it is featured a bit in the discussion).

Validity of the findings

In general, the analyses and results seem to be scientifically sound, but as mentioned before, I recommend the authors to focus the paper more and make a selection of figures and tables. At the moment, it is just overwhelming and it is easy to lose the context.

The discussion and conclusion part needs a major revision, once the aim and objectives of the introduction, method and results parts have been better tailored.

Additional comments

Additional comments for specific sections:

Line 9: specify what the data cube is for →
In line 10 and throughout the paper: spell Earth Observation, instead of earth observation
Line 32: data-technology gap needs a better explanation here. I do not well understand the reasoning in the sentence
The quote in line 34 would need a proper citation, under the link provided I cannot find it and it seems a bit out of place
Line 39-40: would revise or remove this sentence --> 'users cannot lag behind computing capacity'
Line 56: EarthCube.org is not a datacube and Digital Earth Australia and Digital Earth Africa are from the same 'Open Data Cube' initiative
Line 58: EODC is a company, not an EO platform
Line 60: I would remove this sentence
Line 89: why do you only present four datasets and do not list all data the data cube is made of?
Line 100: this is a strong statement which I would rather remove. Be factual in this section and describe what you did and why
Line 108 to 111: this can come in the introduction
Figure 2: why is the period for Sentinel-2 NIR 10m March-December 2018 and not also June-October 2018?
Line 156: why the data from 2000 to 2019 and not up to 2020?
There is no reference to Figure 2, Figure 5 and Table 2 in the text
Line 170: I would remove analysis-readiness in this context
Line 170-174: can go to introduction
Figure 7 is referenced before Figure 6

I do not entirely understand the part of the land cover experiments. Why do you do these experiments?

The definition of the seasons is not clear to me - it seems that winter is defined for months Dec to Mar, but in fact it is defined for the months Dec, Jan, Feb. The same applies for the other seasons.

Reviewer 2 ·

Basic reporting

The paper describes the establishment of an data cube consisting of various earth observation datasets and the methods used to create these. It describes and documents the methodological steps performed.

Experimental design

The research of the paper is within the aims and scope of the journal and fullfills important answer to answer to existing knowledge gaps. The experimental design follows a line from identifying research questions, providing a solution, validating it and showcasing an example of application. The methods are sufficiently described, however a couple of suggestions as outlined in the additional findings should be addressed before publication. The gap filling method should rely on existing established methods.

Validity of the findings

The validity of the findings are fine as the paper mainly focus on the generation of a EO datacube. A comparison against existing scientific literature for some sections (detailed in section 4 blow) would need to be addressed before publication.

Additional comments

L10 '... quarterly to annual estimates ' - Does this also apply to the Digital Elevation Model ? Otherwise please rephrase.
L18-20 Please state reason for observed higher performance over which other methods. As such the sentence can not stay like that.
L22 DTM - what is that ? introduce acronyms first time using it in abstract/paper- Please revise whole manuscript with respect to acronym usage
L23 'combining all four datasets' - please detail the method, otherwise it's unclear to the reader what has been done
L40-41 - Reference please for these statements
L43 'minimize the risk of error propagation' - From a reviewers point of view you can not minimize a risk, you can reduce the error- Same applies for overoptimising decisions - also this is not possible. Please redraft.
L46 - please provide reference after 'challenges'.
L47 replace developers with goals
L52 'perfectly aligned, complete and consistent' - please provide a definition/reference against your work is evaluated. which space-time continuium are you refer too ?
L56 Earthcube.org misses reference
L73 '16 day aggregation format' - unclear what is this thing - 'large' what is large?
L74 'limitations for large scale analysis' - please provide evidence for that
L75 please provide evidence for this statement in form of a table like how many %, significant level tested with which method
L77 please revise the sentence and give evidence - its unclear - why could a user not using ESA/DIAS infrastructure to achieve the same
L95 why are the 10m S2 aggregates are not done at quarterly level ?
Fig1. what is the meaning of 'complete' in complete landsaet quaerterly 30 mosaics' - why is that not in the S2 ARD ? L9 is still flying and producing data every day
L132 The article falls short in view of gap filling methods - at least from the reviewed paper it looks like to the authors have not been taken into account any of the other internationally recognized methods like ML, geostatistical interpolations, DINEOF,..... The paper needs to have a section where the various methods are outlined and the authors need to clearly state why not an already existing method has been applied.
L147-L155 The reviewer question the approach of searching across the whole time range instead of taking the spatial dimension into account. The climate has changed decades significantly and imputing a value from (L149) from any year in the timeseries is questionable. The authors should provide evidence why this is optimal solution (numbers, tests)
L181-182 how is reprojection done - which method, how are cell alignments to LL corner assured across all various cell geometries from input data sources?
L180-184 - where is the codebase for these tasks to test reproducibility ?
Table 4 - please add references for EU-DEM and GLO30 plus resolutions
L234 . what is the impact of a 5 arc degree on the much more detailed DTM cells. Have the authors downsampled/interpolated the 5arcdegree data ?
Placement of tables and figures in the review document is suboptimal - please also review
L243 which gdal version - please review whole manuscript to specify software versions
L221 which software/codebase for reproduction ? R/Python/C/Rust ?
L246 LUCAS reference should be to EUrostat and not to d'Andrimont
L251 which software was used to perform RF
Fig 8 consider making one figure out of it (on legend, one title with a), b),c) and d) instead of having 4 seperate figures. This also applies to the other figures in the manuscript
Fig 11 - A legend entry with 0.0- 0.0 can not exits
L335 please rewrite and discuss at length
Table 11 what is EML DTM ? what are the B values behind the numbers ?
FIgure 12 please discuss difference between landsat and copernicus gap filling for time period 2018-2021.
FIr 13 - why are there higher number of gaps along the boundaries towards the east . DO ethe authors have a processing artifact here as there scene selection was suboptimal ?
L369 'computationally efficient' - compared to what ? as mentioned above (L132) a better approach would have been to use an already existing and validated method.
L382 'time constraints' what reason is that ? Do not understand
L395-401 - really strange (L399) is that a processing artifact by the authors ? Please provide evidence
Fig 18 - which model do the authors use here ?
L452 'posit' ?
L493 include EEA behind environmental monitoring ?

---

## Round 0.2 · accepted · Accept

Thank you for your efforts. This manuscript is ready for publication.

Reviewer 1 ·

Basic reporting

no comment

Experimental design

no comment

Validity of the findings

no comment

Reviewer 2 ·

Basic reporting

Comments have been addressed

Experimental design

no comment

Validity of the findings

no comment